# A Biosurfactant from *Candida bombicola*: Its Synthesis, Characterization, and its Application as a Food Emulsions

**DOI:** 10.3390/foods11040561

**Published:** 2022-02-16

**Authors:** Maria Isabel Silveira Pinto, Jenyffer Medeiros Campos Guerra, Hugo Morais Meira, Leonie Asfora Sarubbo, Juliana Moura de Luna

**Affiliations:** 1Escola Icam Tech, Universidade Católica de Pernambuco (UNICAP), Rua do Príncipe, n. 526, Boa Vista, Recife 50050-900, Brazil; misp13@hotmail.com (M.I.S.P.); hugo.meira@iati.org.br (H.M.M.); leonie.sarubbo@unicap.br (L.A.S.); 2Departamento de Engenharia Química, Universidade Federal de Pernambuco (UFPE), Av. Prof. Moraes Rego, 1235, Cidade Universitária, s/n, Recife 50670-901, Brazil; jenyffer.campos@ufpe.br; 3Instituto Avançado de Tecnologia e Inovação (IATI), Rua Potyra, n. 31, Prado, Recife 50751-310, Brazil; 4Escola de Saúde e Ciências da Vida, Universidade Católica de Pernambuco (UNICAP), Rua do Príncipe, n. 526, Boa Vista, Recife 50050-900, Brazil

**Keywords:** bioemulsifiers, industrial waste products, food additives

## Abstract

The present study aimed to produce a biosurfactant from *Candida* yeast cultivated in a low-cost medium made of sugar-cane molasses (5%), frying oil waste (5%), and corn steep liquor (5%). Initially, the production at the flask-scale was investigated and then scaled up in bioreactors to 1.2, 3.0, and 50 L to simulate a real production scale. The products obtained an excellent reduction in surface tensions from 70 to 29 mN·m^−1^ in the flask-scale, comparable to 33 mN·m^−1^ in the 1.2-L reactor, to 31 mN·m^−1^ in the 3-L reactor, and to 30 mN·m^−1^ in the 50-L reactor. Regarding the yield, it was observed that the isolation by liquid-to-liquid extraction aided biosurfactant production up to 221.9 g·L^−1^ with a critical micellar concentration of 0.5%. The isolated biosurfactant did not exhibit an inhibitory effect on the germination of vegetable seeds and presented no significant acute toxicity in assays with *Artemia salina* and *Allium cepa*. Among the different formulations of mayonnaise-like sauces, the most stable formula was observed with the addition of the biosurfactant at a concentration of 0.5% and the greatest results were associated with the guar and carboxymethyl cellulose gums. Thus, the biosurfactant from *C. bombicola* represents a promising alternative as a food additive in emulsions.

## 1. Introduction

Biosurfactants are well-known for their applicability in the environmental field, primarily in the processes of bioremediation, the removal of toxic metals from the soil, oil and gas processing, and enhanced oil recovery [1,2,3,4,5]. Due to their amphiphilic structures, the biosurfactants are able to increase the surface area of water-insoluble species, enhancing their bioavailability and altering the properties of bacterial cell surfaces, making them excellent emulsifiers, as well as foaming and dispersing agents. Compared to the chemically-synthesized surfactants, they present some general advantages, such as biofriendly production, higher biodegradability, lower toxicity, and potential in a range of biotechnological applications [6]. Byproducts from industrial waste can add value to the production of biosurfactants by composing the substrate, in addition to reducing their pollutant effects when released into the environment [7].

Their physical and chemical properties make biosurfactants attractive in industrial and biotechnological applications, such as in food additives, cosmetics, detergents, agriculture, and medicine [8,9,10]. The emulsifying, foaming, humectant, solubilizing, and anti-adhesive properties of biosurfactants are exceptionally desirable in the food industry [11,12]. The most common use is in stabilizing emulsions in dairy products to create a satisfactory texture and creaminess. Furthermore, they are added to retard agglomeration, solubilize aromatic oils, enhance organoleptic properties in bakery products and ice cream formulations, and stabilize fat while frying [13]. Before being applied in food products, biosurfactants should be evaluated due to their antimicrobial and non-stick properties [14,15,16] and the absence of toxicity in vitro from a perspective of production cost reductions and green surfactant innovations [17,18].

Among the microorganisms able to produce biosurfactants, the genus *Candida* has received great attention in food processing applications, including *C. intermedia, C. maltosa, S. versatilis,* and *C. zeylanoides* [19,20]. Studies related to optimizing biosurfactant production from regional oil substrates and alternative glucose sources pointed out the use of *Candida lipolytica* [21]. Other works employed an association of vegetable oil- and carbohydrate-based substrates for biosurfactant production from *Candida* [22], while industrial waste has emerged as a promising source for substrates [23,24]. In addition, the application of biosurfactants from *Candida bombicola* [25] and *Candida utilis* [26] has gained attention in the cookies industry and in the production of mayonnaise-like sauces [27,28].

Polysaccharides have been widely used in food products to control rheological and organoleptic properties, to modify texture, stabilize emulsions, suspensions, and foams, as well as to inhibit ice and sugar crystallization and control the release of active compounds, such as flavors and antioxidants [29]. Xanthan gum (XG), guar gum (GG), carboxymethyl cellulose (CMC), and starch derivatives are the most common ones in the food industry, especially in emulsion-based products, such as mayonnaise, salad dressings, concentrated fruit beverages [30], and bakery products [31]. Therefore, the search for a natural and non-toxic bioproduct with stabilizing and emulsifying properties has aroused the interest of the scientific community to develop new ingredients and additives for food products. For this reason, a bioemulsifier produced from *Candida bombicola* was investigated as a food additive in this research paper.

## 2. Materials and Methods

### 2.1. The Microorganism and Storage Medium

The *Candida bombicola* strain was kept at 5 °C in a YMA (yeast mold agar) medium to perform the assays. The medium composition in *w/v* was as follows: yeast extract (0.3%), malt extract (0.3%), tryptone (0.5%), D-glucose (1.0%), and agar (5.0%). To maintain the cellular viability of the colony, samples were transferred monthly to new fresh media.

### 2.2. Microorganism Growth

The inoculum growth was carried out in YMB (yeast mold broth), similar to YMA except for the presence of agar. Initially, the inoculum was transferred to a YMA tube in order to obtain a young, standardized colony. Then samples were transferred to Erlenmeyer flasks with 50 mL of YMB in each and were incubated for 24 h at 28 °C at 200 rpm. After the incubation period, the samples were diluted up to a concentration of 10^6^ cells mL^−1^.

### 2.3. Biosurfactant Production

The production of the biosurfactant was carried out by the fermentation of the yeast samples a distilled-water-based media with 5% sugar-cane molasses, 5% frying oil waste, and 5% corn steep liquor. The molasses were provided by Usina São José (Recife, Pernambuco, Brazil), the frying oil from the local food industry, and the corn steep liquor from Corn Products S.A. (Cabo de Santo Agostinho, Pernambuco, Brazil).

### 2.4. Scaled-Up Biosurfactant Production

In addition to the flask production, the bioreactors of 1.2, 3.0, and 50 L (MA502, Marconi LTDA, Piracicaba, Brazil) were employed to investigate the magnification of the process under an orbital shake for 120 h at 28 °C and at 180 rpm. After fermentation, the broth samples were collected for the evaluation of emulsification activity, surface tension, and interfacial tension and, thereafter, to perform the isolation of the biosurfactant.

### 2.5. Biosurfactant Isolation, Purification, and Characterization

To determine the emulsification activity, the samples were centrifuged at 4500 rpm for 15 min and were analyzed according to the Cooper and Goldenberg [32] methods. All the assays were carried out in triplicate.

The surface tension (ST) was obtained with a KSV Sigma 700 tensiometer (Helsinki, Finland) by using the du Noüy ring method. The interfacial tension (IT) was measured against n-hexadecane in the cell-free broth after millipore filtering (0.45 µm). The tension was considered high when it was above the value of 18 mN·m^−1^, and was considered low when the values were below 7 mN·m^−1^. The critical micelle concentration (CMC) of the isolated biosurfactant was determined automatically in the tensiometer [33].

The isolation was performed by both liquid extraction (LE) and acid precipitation (AP) and was compared. For the liquid extraction methodology, ethyl acetate was used as a solvent and the extraction was performed three times in a non-centrifuged broth. The organic phase was then separated and sodium sulfate was used to form the precipitate. Then, the sample was filtered and dried [33]. For the acid precipitation, the broth was centrifuged at 4500 rpm for 20 min. The temperature of the samples was kept at 10 °C to aid cell removal. The pH was adjusted to 2.0 by adding HCl (6.0 M) and was precipitated by adding two volumes of methanol [34].

For purification, the biosurfactant was added to solvents with increasing polarities and was analyzed in thin-layer chromatography on silica gel G60 plates (Merck, Germany). After purification, the biosurfactant was analyzed by infrared spectroscopy and nuclear magnetic resonance (NMR) [35].

### 2.6. Phytotoxicity Assays with Seeds

The phytotoxicity of the biosurfactant was evaluated in static assays to estimate germination rates and the relative root growth of *Solanum lycopersicum* (tomato) and *Cucumis anguria* (gherkin), as described by Tiquia et al. (1996) [36]. The isolated biosurfactant was prepared in distilled water at different concentrations (1/2 CMC, 1 CMC, and 2 CMC). The assays were performed in triplicate for 5 days in the absence of light. At the end of the experiments, the relative seed germination (RSG), relative root growth (RRG) (≥ 5 mm), and the germination index (GI) rates were calculated as in Equations (1)–(3), respectively.
(1)$$RSG\;\left(\%\right)=\frac{germinated\;seeds\;in\;contact\;with\;the\;samples\;}{germinated\;seeds\;from\;control\;sample}*100$$
(2)$$RRG\;\left(\%\right)=\frac{average\;growth\;of\;roots\;in\;contact\;with\;the\;samples\;}{average\;growth\;of\;roots\;from\;control\;sample}*100$$
(3)$$\;\;\;\;\;\;\;\;\;IG\;\left(\%\right)=\frac{RSG\times RRG}{100}$$

### 2.7. Toxicity Assay with Artemia salina

The toxicity of the biosurfactant was also evaluated against brine shrimp eggs. The larvae were incubated for 24 h before use. Biosurfactant samples were prepared in marine synthetic samples (33 mg. L^−1^) at three concentrations (1/2 CMC, 1 CMC, and 2 CMC). The assays were carried out in 10-milliliter penicillin flasks with 10 larvae and 5 mL of each sample for 24 h and the living organisms were counted [37]. The control was prepared with a marine synthetic sample with no biosurfactant. The threshold of toxic concentration was set as the lower concentration able to kill the organisms within 24 h. All the assays were carried out in triplicate.

### 2.8. Phytotoxicity Assays in Onions (Allium cepa L.)

*Allium cepa L.* specimens were also used as an indicator of toxicity. The biosurfactant samples were also prepared at 1/2 CMC, 1 CMC, and 2 CMC. The inhibition of root growth was observed regarding the exposure to samples as described by Jardim [38]. The assays were performed in triplicate.

### 2.9. Biosurfactant as a Food Additive

The emulsifying property of the product was assessed regarding a mayonnaise-like sauce with the following composition: 40% corn oil (Bunge, Recife, Brasil), 40.3% water, 10% vinegar, 4% powder egg (Naturovos LTDA, Salvador do Sul/RS Brasil), 2% sugar, 2% salt, 1% mustard flour, and 0.5% instant starch (Unilever LTDA, Recife, Brasil) [39]. To this formulation, four thickening agents were investigated: Arabic gum, xanthan gum, guar gum, and carboxymethylcellulose at amounts of 0.2, 0.5, and 0.8%, until the most consistent sauce was obtained. The pH of each sample was measured and the viscosity of each formulation was determined at 27 °C by using a rotary viscometer (NDJ-1, Brookfield).

Once the best conditions were set, the formulation was elaborated with the isolated biosurfactant at concentrations ranging from 0.2 to 0.8% to obtain the most stable emulsion. The samples were stored at 4 °C for 6 months and then they were visually inspected [40].

### 2.10. Microbiological Analyses

All the samples were evaluated after their 6-month refrigeration at 4 °C (± 2 °C) regarding microbiological parameters. The analyses were carried out in the Laboratory of Animal Source Foods at the Federal University of Pernambuco, according to the methods recommended by AOAC (2005) [41]. The investigated parameters were in accordance with the current Brazilian legislations: the determination of the most probable number (MPN) of total and thermotolerant coliforms, the *Staphylococcus aureus* count, and *Salmonella* sp. presence [42].

## 3. Results and Discussion

### 3.1. Biosurfactant Properties

The CMC is the minimum concentration of a biosurfactant necessary for the maximum reduction in the surface tension of water and the onset of the formation of micelles. This concentration is used as a measure of the efficiency of a biosurfactant. The biosurfactant produced by *C. bombicola* had a CMC of 0.5%, which falls within the range that is considered promising for the production of a biosurfactant.

The biosurfactant from *C. bombicola* exhibited low interfacial tension (3.5 mN·m^−1^), which indicates the facility in forming stable emulsions. The reduction in surface or interfacial tension is considered one of the main parameters for the detection of surfactant presence [43]. In addition to this, the stability of oil/water emulsions is also relevant as a parameter because the ability of a molecule to form a stable emulsion is not always linked to a reduction in surface tension [44]. Luna et al. [45] found an interfacial tension of 12.5 mN·m^−1^ for a biosurfactant from *Candida sphaerica* cultivated in an industrial waste-based broth.

The use of industrial residues can reduce the cost of biotechnological processes by approximately 30% [44]. In the present study, the biosurfactant from *C. bombicola* was obtained by its cultivation in distilled water, supplemented with 5% canola frying oil, 5% molasses, and 5% corn steep liquor.

The use of bioreactors stands out among the strategies that can be used to increase biosurfactant production yields, as such equipment, which constitutes a completely closed system that enables the control of emissions, the greater control over the different variables of the process (pH, temperature, humidity, etc.), the better incorporation of additives, and a reduction in processing time, all of which are fundamental aspects in industrial applications [44].

Table 1 exhibits the estimated properties of the biosurfactants produced in flask and bioreactor scales.

As can be seen in Table 1, the biosurfactant exhibited a great reduction in ST when compared to water (70 mN·m^−1^) with the greatest reduction in the flask scale and an increasing reduction when the bioreactor is magnified. The use of different oils presented a minor influence regarding the emulsification activity, with an average index of 47% for the flask-scale process and 30, 11, and 58% in 1.2-, 3.0-, and 50-L bioreactors, respectively. LE was demonstrated to be more effective for isolating the biosurfactant and the scale-up was notably favorable to the process, allowing the obtainment of up to 221.9 g·L^−1^ of product.

Rau et al. [46] found yields of sophorolipids up to 300 g·L^−1^ using *C. bombicola* ATCC 22,214 cultivated in a bioreactor with waste and glucose. Almeida et al. [2] employed *C. tropicallis* grown in 2.5% of molasses and 2.5% of waste canola oil in the production of a biosurfactant. The surface tension was equivalent to 29.52 mN·m^−1^ with a yield of up to 7.0 g·L^−1^. Luna et al. [45] grew *C. sphaerica* in a medium made of 9.0% soya oil and 9.0% corn steep liquor and the surface tension was equivalent to 25 mN·m^−1^ and the yield was estimated at 8.0 g·L^−1^.

Marti et al. [47] described the production of surfactin from genetically modified strains of *Bacillus subtilis,* a mineral medium containing 2% glucose in shaker flasks and a 5-L bioreactor. In contrast to the present results, surfactant production was 6.2 g·L^−1^ in flasks but did not surpass 0.006 g/L in the bioreactor. These findings demonstrate the production capacity of *C. bombicola* and its efficiency regarding different production volumes in comparison to *Bacillus subtilis.*

Marti et al. (2014) described the production of surfactin from genetically modified stains of *Bacillus subtilis* in a mineral medium containing 2% glucose in shaker flasks and a 5-L bioreactor. In contrast to the present results, surfactant production was 6.2 g/L in flasks, but did not surpass 0.006 g/L in the bioreactor. Growth kinetics of *Candia tropicalis* and biosurfactant production

### 3.2. Biosurfactant Characterization

The H^1^-NMR and C^13^-NMR spectra for the biosurfactant are shown in Figure 1 and Figure 2, respectively.

In Figure 1, it is possible to observe a signal in the region below 1 ppm, which can be associated with the (CH_3_)_n_ group, and peaks between 1 and 2 ppm, which corresponds to the signal of (H_2_C-CH_2_)_n_ e (CH_2_)_n_. The characteristic peaks in the range of 2 and 3 ppm indicate the presence of (HC=CH-CH_2_)_n_ e (CH_2_)_n_ groups. The signal observed at around 5.4 was associated with the (HC=CH)_n_ group in the structure of the biosurfactant. Therefore, hydrogen bonds for aliphatic carbon and sp^2^-hybridized carbon can be noted in the spectrum. A weak peak in the range of 7 and 8 ppm may be indicating the presence of the carboxyl acid group.

According to Figure 2, there is a signal around the 180-ppm region, which corroborates this biosurfactant as a carboxylic acid compound. The two sharp peaks at around 130 ppm indicate the carbon double-bonds, and the aliphatic carbon may be represented by the signals in the 10 to 40 ppm range. These results suggest that the biomolecule of the surfactant is a type of carboxylic acid metabolite, likely bonded to carbohydrates (simple fatty acids), as described for other glycolipid biosurfactants produced by yeasts. Additionally, it is possible to observe a triplet of sharp peaks in the 80-ppm region that may be related to the solvent.

According to Ribeiro et al. [26], *Candida utilis* produces a biosurfactant structure of a metabolized fatty acid rich in oleic acid, which points out a variety of fatty acid proportions. Other reports in the literature described biosurfactants with a glycolipid nature, such as the product of *Bacillus* sp. IITD106 [48]. FTIR and NMR analyses found a saponin structure with two sugar groups and a 5-ring triterpene sapogenin unit. Santos et al. [21] reported promising results by growing *C. lipolytica* in media based on a waste rich in animal fat. The characterization of the biosurfactant suggested its glycolipid structure. According to Figure 2, similar results were found by Campos et al. [49] with surfactant from *Candida utilis*.

### 3.3. Phytotoxicity Assays with Seeds

As was mentioned before, biosurfactants are expected to present low toxic effects. Even though toxicity assays are recommended to guarantee its safe application as an ingredient in food formulation, phytotoxicity assays are of easy operation and execution, fast, and have a low cost, which is highly regarded for scientific research. The germination index (GI) takes into consideration the relative germination measure of seeds and the relative unidimensional growth of roots to estimate the acute toxicity in the development of seeds, such as tomato and gherkin seeds. No inhibitory effects could be noted in seed germination rates and the growth of roots for both tomato and gherkin seeds. The elongation of roots was significant even at high concentrations of biosurfactant and ½ CMC, 1 CMC, and 2 CMC. For the biosurfactant isolated by acid precipitation, IG was estimated at 85, 38, and 22% for the tomato and 100, 100, and 80% for the gherkin, respectively. For the samples isolated by liquid extraction, germination index results of 64, 45, and 0% of were observed for tomatoes, while 41, 28, and 0% were observed for gherkins. In addition, secondary roots were observed to grow in the assays with the gherkin for isolation by liquid extraction. The experiments demonstrated a downtrend in seed germination rates when the biosurfactant was enhanced, as is shown in Table 2.

Table 2 demonstrates that the GI was reported to be lower than 80% for biosurfactants at ½ CMC produced from *Candida lipolytica* UCP0988 grown in a medium containing 5% animal source fat and 2.5% corn steep liquor, according to Santos et al. [21]. On the other hand, Silva [50] found no significant inhibitory effect in the germination rates and root growth of lettuce, cabbage, and coriander seeds for a biosurfactant produced from the yeast 9II. Phytotoxicity against cabbage (*B. oleracea*) was investigated for another biosurfactant sample from *Pseudomonas sp* by Silva et al. [51]. The authors also found the absence of significant toxicity for seed germination and root elongation.

### 3.4. Toxicity Assay with Artemia salina

The toxicity against aquatic organisms is relevant in the case of any application the product may have in aquatic ecosystems. *Artemia salina* is a standard marine living being commonly employed in ecotoxicology due to the feasibility of maintenance on a lab scale, simple growth conditions, and a short life cycle. The assays demonstrated that the AP isolated biosurfactant caused 50 and 100% lethality rates when at high concentrations of 0.5% (CMC) and 1% (2 CMC), respectively. The LE isolated biosurfactant did not exhibit significant lethal rates.

Santos et al. [21] found no significant lethal rates for *Artemia salina* with biosurfactant samples at 0.02 and 0.06%, while 100% of the larvae were found to be dead at 0.08%. A report from Santos [52] exhibited no significant lethal rates for a biosurfactant derived from *Streptomyces* sp. DPUA1559 at amounts of 50, 100, and 150 mg·mL^−1^ in CMC (10 mg·mL^−1^). The biosurfactant produced from *Pseudomonas aeruginosa* exhibited lethal rates at 100 and 50% on the organisms at concentrations of 700 and 525 mg·L^−1^, respectively [36]. Lower concentrations presented no significant lethal rates and the same was observed for the cell-free broth.

### 3.5. Phytotoxicity Assays in Onions (Allium cepa L.)

Plants, such as onions, have been widely explored for the ecotoxicology assessment of several pollutants [53]. The advantages of this vegetal organism are related to its low cost, easy growth, non-seasonal availability, and feasibility to both acute and chronic toxicity assays under laboratory or environmental conditions [54]. The phytotoxicity, estimated by the inhibition of root elongation on mature organisms or seed germination, is the most common indicator [55]. The experiments are fast and easily executed. The toxic effect was regarded from the weight gain and the elongation of the roots at different concentrations of the biosurfactant.

The isolation by LE demonstrated no significant difference regarding root length and weight gain in the investigated concentration range. The product, from AP isolation, displayed a downtrend of growth when the concentration was elevated, and both results were compared with the control samples. The surfactant Tween 80, as reported by Grippa et al. [56], did not exhibit cytotoxic effects over *Allium cepa* after 72 h of exposure. Therefore, toxicology has acted as a powerful tool to estimate the potential risks to the ecosystem and to identify compounds able to compromise the metabolism of natural occurrence biota [21].

### 3.6. Biosurfactant as a Food Additive

#### 3.6.1. Selection of the Thickening Agent

The Arabic gum, xanthan gum, guar gum, and carboxymethylcellulose presented different results in a previous evaluation after 30 days. All the samples with Arabic gum exhibited a thin character and the presence of two phases after 15 days. Xanthan-derived formulations presented thick samples with serum formation in the 0.8% sample. For guar gum and carboxymethylcellulose, no alteration could be observed within 4 weeks. It was possible to note the significant increase in dynamic viscosity when the concentration of the thickening agent was also increased, especially for guar gum and xanthan gum. For Arabic gum, the dynamic viscosity did not present a strong relation with the enhancement of its concentration in the formula.

Xanthan gum is an anionic gum commonly seen in oil/water emulsions, such as dressings and mayonnaise-based sauces, to achieve a jellified texture. It is highly pseudoplastic and remarkably stable regarding acidic conditions, temperature, and enzymes [30]. Guar gum is made of polysaccharides with a high molecular weight and can be found in many applications in the food industry. When incorporated at around 1 g for each 100 g, it is able to provide gelation, emulsification, and thickening properties to food products. At superior rates, it presents high viscosity and limits the application of these types of products [57].

Chivero et al. [58] evaluated soy soluble polysaccharides, octenyl succinate starch, and Arabic gum in the production of mayonnaise and related a similar result for Arabic gum, especially when oil composition exceeded 60% *v*/*v*. Martín-Alfonso et al. [59] studied the rheological properties of aqueous solutions containing guar gum and xanthan gum, ranging between 1–3% (*w*/*w*), and observed that the rheokinetics of the process, and the resulting rheological responses, were extensively altered by the hydrocolloids. This suggests that xanthan solutions behaved as weak gels, whereas the entanglement and the formation of a viscoelastic gel-like structure were referred to the guar gum samples. Bak and Yoo [60] observed a synergic effect of viscosity in xanthan and guar gums, only in the presence of sodium chloride and sucrose.

Regarding the pH of samples, the average values were around 3.70 and the lowest average was obtained with guar gum at 6.63. All the products are under the recommendation for mayonnaise formulas. The pH-acceptable range is from 3.3 to 3.8. The results are displayed in Table 3.

Nanoemulsions, produced by the addition of rhamnolipids, were unstable at around pH 4.0 when a thin layer of oil over the surface was formed after storage. The authors concluded that the electrostatic repulsion was not strong enough to overcome any attractive interactions (e.g., van der Waals) acting between the droplets, thereby leading to droplet aggregation at pH 2 and 3 [61]. Xanthan gum can keep viscosity constant within a wide range of temperatures and pH, forming high viscous samples when at low concentrations [30]. Emulsions with surfactin, on the other hand, remained stable only when the pH was between 6.0 and 9.0, because lower pH conditions, which are easily found in food, cause the precipitation of the aspartate and glutamate acid byproducts, as stated by Hoffmann et al. [62]. It suggests the low applicability of this last compound in food emulsions.

#### 3.6.2. Biosurfactant Isolation Influence on Emulsion Stability

Since guar gum and carboxymethylcellulose remained stable after 30 days, they were selected for the investigation of the influence of the biosurfactant isolation method on the stability of the emulsions. The stability parameters were evaluated according to the absence of serum and the separation of phases when the concentration of the biosurfactant was varied. The collected data is presented in Table 4 and the ‘+’ sign was attributed to two-phase samples while the ‘−’ sign represented visually stable emulsions.

As can be seen in Table 4, when the biosurfactant is at 0.5% *v/v*, all the products exhibited the desirable stability of emulsions. However, the serum was observed for three out of the four formulations when the concentration or biosurfactant was at 0.3%. The concentration of the biosurfactant was then selected at 0.5% for further analyses.

Campos et al. [28] investigated the manufacturing of mayonnaise-like sauces with a biosurfactant from *Candida utilis* in the presence and absence of thickening agents. The authors reported the best results when guar gum was associated with 0.7% of biosurfactant, preserving the emulsion after 30 days. Chen et al. [63] employed saponin, which ranged in concentration from 1.5–3.0% (*w/v*) in ultrasonic emulsification, and they found out that the higher the surfactant concentration, the larger the interfacial area, which lowered the liquid interfacial tension compared to other emulsifiers, even after 30 days of storage.

### 3.7. Microbiological Analyses

None of the samples presented microbial contamination after 180 days of storage. According to the results, the absence of *Salmonella* sp./25 g was noted, the coagulase-positive *staphylococci* levels were below 10 CFU·g^−1^, and the coliforms below 3.0 MPN·g^−1^ at 45 °C. The microbiological analyses were in accordance with the mayonnaise-like emulsion presented by Campos et al. [49] with a biosurfactant from *Candida utilis*. These results corroborate the premises that all the steps of the production process were monitored from the selection to the use of ingredients and with respect to good manufacturing practices to prevent any risks to the health of consumers.

## 4. Conclusions

In the present work, the production of biosurfactant by *Candida bombicola* was attained in the laboratory at a large scale using a low-cost growth medium. The product presented surfactant and emulsifying properties and a desirable yield. The scale-up in bioreactors favored the process for future industrial applications. The isolated biosurfactant exhibited no toxicity, which shows it is safe to incorporate in sauces and dressings due to its attractive physical, chemical, and textural properties. The success in the properties of a mayonnaise-like sauce could be observed. Thus, the biosurfactant demonstrated promising characteristics to act as an emulsifying additive in food products.

## Figures and Tables

**Figure 1 foods-11-00561-f001:**
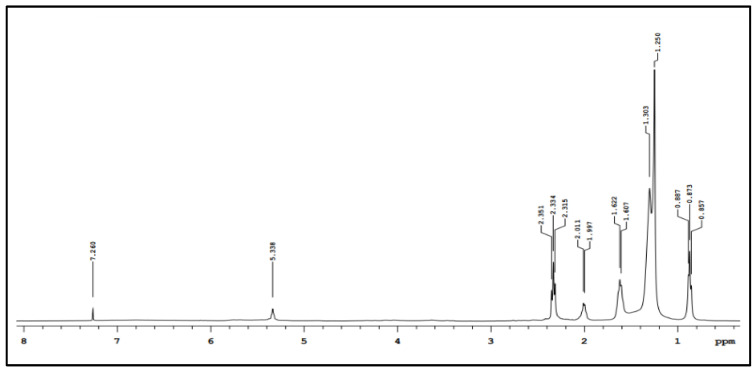
Hydrogen nuclear magnetic resonance spectrum of biosurfactant produced from *C. bombicola* growth.

**Figure 2 foods-11-00561-f002:**
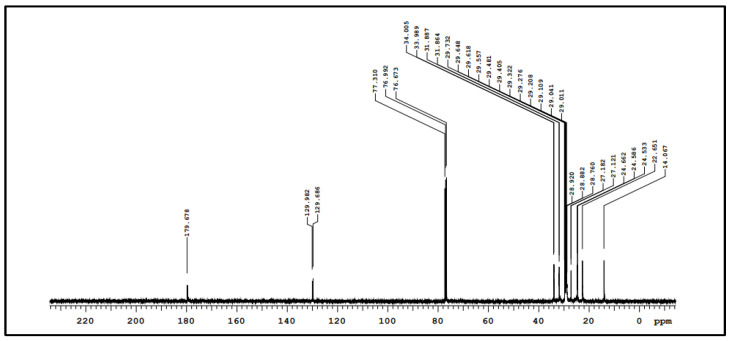
Carbon nuclear magnetic resonance spectrum of biosurfactant produced from *C. bombicola* growth.

**Table 1 foods-11-00561-t001:** Biosurfactant properties when produced in flask scale and 1.2-, 3.0- and 50-L bioreactors.

Method	ST ^a^ (mN·m^−1^)	Emulsification Index (%)			Yield (g·L^−1^) ^b^	
		Canola Oil	Corn Oil	Soya Oil	LE Isolation	AP Isolation
Flask	29.0 ± 1.0	49.0 ± 1.2	47.0 ± 0.9	45.0 ± 1.1	12.5 ± 1.4	5.9 ± 1.3
1.2-L bioreactor	31.0 ± 1.1	29.0 ± 1.4	32.0 ± 0.9	29.0 ± 1.1	19.8 ± 0.9	2.8 ± 1.0
3.0-L bioreactor	33.0 ± 1.3	10.0 ± 1.2	10.0 ± 1.1	12.0 ± 1.6	61.0 ± 0.8	1.1 ± 1.1
50-L bioreactor	30.0 ± 1.3	59.0 ± 0.9	58.0 ± 1.1	57.0 ± 1.2	221.9 ± 1.1	2.4 ± 1.3

^a^ ST = surface tension; ^b^ LE = liquid-extraction; AC = acid precipitation.

**Table 2 foods-11-00561-t002:** Toxicity assays of biosurfactant from *C. bombicola* grown in 5% waste oil, 5% molasses, and 5% corn steep liquor with tomato and gherkin seeds.

	Germination Index of Isolated Biosurfactants (%)					
	½ CMC		CMC		2 CMC	
Seeds	LE ^a^	AP ^b^	LE	AP	LE	AP
*Cucumis anguria* (gherkin)	100% ± 0.1	41.0%± 0.1	100% ± 0.1	28.0% ± 0.1	80.0% ± 0.2	0% ± 0.1
*Solanum lycopersicum* (tomato)	85.0% ± 0.1	64.0% ± 0.1	38.0% ± 0.1	45.0% ± 0.2	22.0% ± 0.2	0% ± 0.1

^a^ LE = biosurfactant isolated by iquid extraction; ^b^ AP = biosurfactant isolated by acid precipitation.

**Table 3 foods-11-00561-t003:** pH measurement for mayonnaise-like sauces produced with isolated biosurfactant from *Candida bombicola* and different thickening agents.

Thickening Agent (% *p/p*)		pH
Xanthan gum	0.2	3.70
	0.5	3.70
	0.8	3.71
Guar gum	0.2	3.56
	0.5	3.63
	0.8	3.70
Carboxymethylcellulose	0.2	3.78
	0.5	3.76
	0.8	3.66
Arabic Gum	0.2	3.71
	0.5	3.71
	0.8	3.71

**Table 4 foods-11-00561-t004:** Presence and absence of phase separation in mayonnaise-like sauces with guar gum and CMC according to the concentration of biosurfactant isolated by LE and AP.

Biosurfactant Concentration (% *v/v*)	Guar Gum		Carboxymethylcellulose	
	LE Isolation	AP Isolation	LE Isolation	AP Isolation
0.2	+	−	−	+
0.3	+	+	+	−
0.4	+	+	−	+
0.5	−	−	−	−
0.6	−	−	−	+
0.7	−	+	+	−
0.8	+	+	−	−

## Data Availability

Data is contained within the article.

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
