# Peer review of "A Biosurfactant from Candida bombicola: Its Synthesis, Characterization, and its Application as a Food Emulsions"

_foods, 2022, doi:10.3390/foods11040561_

Round 1

Reviewer 1 Report

Review of manuscript 1557275 for “Foods” journal

The reviewer believes that this is a manuscript dealing with a very interesting and actual topic of both technological and industrial interest concerning Synthesis, Characterization and food Application aspects of Biosurfactant From Candida bombicola. The authors have considered the most relevant and updated literature evidence in this scientific area and present their results in meaningful flow. However, in order to enable its publication the paper should undergo a heavy quality and linguistic check along with some elaboration of certain parts. At present, it is missing clarity in particular in the discussion section. A few indicative comments are given below:

 Comments per section:

 Abstract

Line-16: The sentence needs some reformulation.. see below suggestion in track:

“…The aim of the present study was the  production of bio-surfactant cultivated in a low- 16 cost medium made…”

 Introduction

 Line-40: perhaps to replace “feasible” with “trigger”?

 Line-68: perhaps you mean “emulsion-based products” ?

I would also suggest to add a small paragraph at the end of this section to explain the originality/novelty of this manuscript…Is the presented experimental work of added value in this scientific field and how?

 Materials and methods 

  • In paragraphs 2.5 and 2.7 there should be reference to literature sources describing the described methods. Unless the protocols have been prepared in this lab…if so this should be explicitly indicated…
  • At the end of the section, please add a short paragraph to describe the statistical analysis performed in the presented results of section 3.

 Discussion

Please check again this section to improve clarity of the presented arguments and explanations. For instance, it is rather difficult to follow the flow of the results interpretation in the following parts:

  • Lines 291-294
  • Lines 303-306
  • Lines 313-316
  • Lines 342-346

Please improve English/split text in smaller sentences/ensure that your text is comprehensive for the reader

 Author Response

Dear Reviewer,

We appreciate your consideration of our manuscript and look forward to having responded to your suggestions. If you have any queries, please do not hesitate to let me know.

Yours sincerel

Reviewer 2 Report

There is a need to improve the english as some of the sentences does not flow well and syntax error. 

Please provide reference for each method of experiment. 

2.9 Is it a viscometer?

Table 1 and 2- provide mean plus minus SD and sig difference 

Table 3. The average and SD should be for each % of thickening agent and not combined. Any rationale to combine?

A lot of scientific name not italic. Also, please follow SI unit and check the -1 (superscript) for some. 

To add in the link between the seed toxicity, and also the application in sauces (why is there description of mayonnaise like for sauce?) as food model- do include the rational of all these in introduction to justify the rational in the work flow and experimental design. 

Author Response

Dear Reviewer,

We appreciate your consideration of our manuscript and look forward to having responded to your suggestions. If you have any queries, please do not hesitate to let me know.

Yours sincerel

Round 2

Reviewer 1 Report

For me the manuscript was improved in line to review's recommendation and be ready for publication